# Oxidative Stress in Traumatic Brain Injury

**DOI:** 10.3390/ijms232113000

**Published:** 2022-10-27

**Authors:** Arman Fesharaki-Zadeh

**Affiliations:** Yale School of Medicine, Department of Neurology, Yale University, New Haven, CT 06510, USA; arman.fesharaki@yale.edu

**Keywords:** TBI, oxidative stress, reactive oxygen species, reactive nitrogen species, synapse, inflammation, antioxidants

## Abstract

Traumatic Brain Injury (TBI) remains a major cause of disability worldwide. It involves a complex neurometabolic cascade, including oxidative stress. The products of this manuscript is examining the underlying pathophysiological mechanism, including reactive oxygen species (ROS) and reactive nitrogen species (RNS). This process in turn leads to secondary injury cascade, which includes lipid peroxidation products. These reactions ultimately play a key role in chronic inflammation and synaptic dysfunction in a synergistic fashion. Although there are no FDA approved antioxidant therapy for TBI, there is a number of antioxidant therapies that have been tested and include free radical scavengers, activators of antioxidant systems, inhibitors of free radical generating enzymes, and antioxidant enzymes. Antioxidant therapies have led to cognitive and functional recovery post TBI, and they offer a promising treatment option for patients recovering from TBI. Current major challenges in treatment of TBI symptoms include heterogenous nature of injury, as well as access to timely treatment post injury. The inherent benefits of antioxidant therapies include minimally reported side effects, and relative ease of use in the clinical setting. The current review also provides a highlight of the more studied anti-oxidant regimen with applicability for TBI treatment with potential use in the real clinical setting.

## 1. Introduction

Traumatic brain injury (TBI) has become a significant cause of morbidity and mortality within the United States and worldwide [1,2]. Approximately 1.7 million individuals experience a TBI, resulting in 52,000 deaths annually. A substantial portion of those injured, estimated to be 124,000, develop chronic disability [1,2,3]. The annual cost of disability has been estimated to be $56 billion a year, creating a significant socioeconomic challenge [4]. Depending on a variety of factors including the severity of the injury, age, and premorbid general health, TBI could lead to wide-ranging neurocognitive, neuroendocrine, and neuropsychiatric sequelae [5,6].

TBI could be conceptualized in two different stages, involving a primary acute process and a subacute/chronic secondary injury neurometabolic cascade. The primary injury is typically due to a mechanical anatomical lesion, involving laceration, contusion, intracranial hemorrhage, and diffuse axonal injury [7]. The secondary injury is a delayed and prolonged complex response, leading to long-term neuropathological changes including metabolic alterations, oxidative stress, neuroinflammation, axonal injury, neurovascular changes, and ultimate neurodegenerative changes, depending on the severity of the injury [8,9]. The brain uses approximately 20% of the body’s total amount of oxygen and demands the highest oxygen supply in the body. An elevated oxygen consumption rate post-injury heightens the probability of producing reactive oxygen species (ROS).

The elevated metabolic rate heightens the probability of the production of reactive oxygen species (ROS). The brain parenchyma also has significant amounts of unsaturated fatty acids and polyunsaturated fatty acids (PUFAs) that are especially vulnerable to free radical reactions and lipid peroxidation. The brain also contains high quantities of iron that can be directly involved with free radical injury [4,10]. The brain also has limited defense mechanisms available against oxidative stress based on its catalase supply compared to other organs such as the liver [11].

This review aims to examine the underlying pathophysiological mechanisms leading to TBI-induced oxidative stress and resultant chronic neuroinflammatory sequelae. Oxidative stress is a product of the orchestration of multiple cellular processes including the activation of oxidative enzymes, nitric oxides, and lipid peroxidation, as well as enhanced excitotoxic activation of glutamatergic neurons [1,12,13]. The resultant oxidative process would ultimately lead to the chronic sequelae of neuroinflammation and synaptopathy. In addition, the discussion will include a number of robust antioxidant regimens reported in preclinical and clinical studies [13].

## 2. Pathophysiology of Oxidative Stress in TBI

The pathophysiological etiology of TBI is heterogenous and occurs on multiple scales. TBI-induced pathological changes include axonal shearing otherwise known as diffuse axonal injury (DAI), neurovascular compartment damage, and neuronal network disruption, as well as apoptotic and necrotic cellular damage [14,15]. The neuroinflammatory response post-injury involves pro-inflammatory cytokines and free radical formation [16], further elaborated in Section 4 below. Oxidative stress plays a key role in this process, and it involves the production of reactive oxygen species (ROS) and reactive nitrogen species (RNS) as outlined in Figure 1. Oxidative stress is a likely result of disequilibrium between the biochemical processes leading to the production of ROS, and the processes responsible for the removal of ROS, involving the enzymatic and non-enzymatic anti-oxidant cellular defense system [17] listed in Table 1. Oxidative stress has been proposed to lead to the formation of brain edema due to mitochondrial dysfunction, blood–brain barrier (BBB) breakdown, sensory-motor dysfunction, and secondary neuronal injury [18,19,20,21]. This review article examines a range of both preclinical and clinical TBI studies with a focus on the central role of oxidative stress in TBI pathology, as well as potential treatment modalities with real translational clinical applications. There are a number of preclinical TBI models proposed, which include the fluid percussion model, the controlled cortical impact model, Feeney’s weight drop model, Marmarou’s weight drop model, the penetrating ballistic-like brain injury (PBBI) model, and the blast wave model [22,23].

### 2.1. Oxidative Enzymes

Oxidative stress is a product of the disruption of the equilibrium between the production of oxygen-derived free radicals and the scavenging antioxidant system. The oxidative agents include hydrogen peroxide (H), its high oxidative metabolic demand, limited antioxidant capacity, and repair mechanisms [39].

ROS are products of arachidonic acid (AA) cascade activity, mitochondrial leakage, catecholamine oxidation, neutrophils, as well as oxidative stress-inducing enzymes [40]. The oxidative stress-inducing enzymes include NADPH oxidase (NOX). NOX is a multi-component enzyme, which consists of cytoplasmic subunits, which include p47phox, p67phox, p40phox, and Rac2. Once phosphorylated, the NOX enzyme can form a more complex structure, translocate to the plasma membrane, and dock with plasma membrane subunits p91phox and p22phox. NOX catalysis occurs at the p91phox subunit (NOX2). Prior studies have reported that overactivation of NOX2 leads to oxidative damage in ischemic, traumatic, and neurodegenerative conditions [7,18,41].

### 2.2. ROS and Oxidative Stress

ROS- superoxide (O2●−), hydrogen peroxide (H_2_O_2_), peroxyl (ROO^−^) radical, and the hydroxyl radical (●OH) are the products of 1-electron reduction of oxygen [1]. These are produced by mitochondria and other cellular enzymes as products of cellular oxidative metabolism. These radicals can, in turn, react with lipids, nucleic acids, and proteins, and lead to downstream protein inactivation including enzymes, receptors, and ion channels [24]. With iron present, superoxide and H_2_O_2_ can be converted into a more potent oxidative agent, hydroxyl radical. Furthermore, superoxide will react with nitric oxide to generate peroxynitrite (ONOO^−^). Myeloperoxidase (MPO) produces hypochlorous acid (HOCl−) and inducible nitric oxide synthase (iNOS) produces nitric oxide (NO●). Cyclooxygenases (COX-1 and COX-2) produce prostaglandin intermediates from arachidonic acid through peroxidase activity first to PGG2 and then to PGH2. NADPH oxidase (NOX) enzymes catalyze the production of O2●− and H_2_O_2_ from oxygen. These are the major agents of cellular generation of ROS in the cells that can contribute to neuroinflammation [1].

### 2.3. RNS and Oxidative Stress

Reactive nitrogen species (RNS) include various nitric oxide compounds, such as peroxynitrite (ONOO^−^) and nitrogen dioxide (NO_2_). The RNS species generally have longer half-lives than O2●− and ●OH [12,25]. It is important to add that multiple enzymes are involved in the free radical formation, which include nitric oxide synthase (iNOS), endothelial nitric oxide synthase (eNOS), cytochrome P450 (CYP450), cyclooxygenase (COX), lipoxygenase (LOX), and xanthine oxidase (XO) [26,27,42,43]. One of the main species leading to oxidative tissue injury is peroxynitrites (PN; ONOO-). PN is produced by the combination of NO synthases (NOS)-generated NO radical and O2●−. PN can also react with carbon dioxide (CO_2_) and form nitoperoxocarbonate (ONOOCO_2_). Each of the PN-derived radicals can cause lipid peroxidation (LP) cellular damage via the transfer of an electron from a hydrogen atom bound to allylic carbon in PUFAs or lead to protein carbonylation due to reacting with particular amino acids [8]. PN is implicated in TBI pathophysiology in multiple ways. The three NOS isoforms (endothelial, neuronal, and inducible) are upregulated during the first 24 h post-TBI, as reported in preclinical studies [21,44]. In addition, prior reports have shown that acute treatment with NOS inhibitors, such as L-NAME, can have a neuroprotective effect and lead to an improved neurological outcome [8,45]. Furthermore, the biochemical marker of PN-mediated damage, based on increased 3-NT levels and ADP ribosylation, has been documented in rodent TBI studies [8,28].

### 2.4. Iron and Hydroxyl Radicals

Unbound iron (Fe^2+^) is one of the most abundant ions within the CNS. Iron dyshomeostasis is a major factor involved in various neurodegenerative conditions including Alzheimer’s Disease (AD), Parkinson’s Disease (PD), amyotrophic lateral sclerosis (ALS), Huntington’s disease, and Friedreich’s ataxia [29]. The iron distribution varies in concordance with the sensitivity of various anatomical regions to oxidative stress [8,46]. Under basal physiological conditions, low molecular forms of redox-active iron are maintained at very low levels. The iron transport protein transferrin tightly binds iron in the Fe^3+^ oxidation state. This ion is stored intracellularly by the iron storage protein ferritin. Post TBI, the pH within injured areas is lowered, in turn leading to the release of iron from its storage sites [8,46,47,48]. The second source of catalytically active iron is hemoglobin. It has been reported that iron released from hemoglobin is responsible for hemoglobin-mediated oxidative damage [48]. Free iron, Fe^2+^, can in turn undergo the following two reactions: First, autooxidation to form Fe^3+^ leading to the formation of O2●− (Equation (1)), or the Fenton reaction to form ●OH (Equation (2)):Fe ^2+^ + O_2_ ---> Fe ^3+^ + O_2_●−(1)
Fe ^2+^ + H_2_O_2_ ---> Fe ^3+^ + ●OH + OH^−^ (Fenton reaction)(2)

It is hypothesized that inflammatory cells that are activated post-injury in TBI are mainly responsible for iron deposition in TBI [29,49]. The dysregulation of ceruloplasmin, the main copper-binding protein, can lead to an increase in free copper ions and a decrease in ferroxidase activity, and, in turn, reduce iron clearance from the CNS. Ceruloplasmin-deficient mice subjected to TBI have been reported to have elevated brain iron levels and enhanced infarct volume [29]. Iron chelation therapies, including deferoxamine and deferipone, have been extensively studied as a treatment strategy in intracerebral hemorrhage (ICH) and stroke [30,31], and are a promising potential regimen in TBI [29].

### 2.5. Lipid Peroxidation

The phenomenon of lipid peroxidation (LP) has been reported in stroke studies [8]. LP has been associated with the loss of cell membrane integrity, increased membrane permeability, and diminished activity in membrane ATPase, leading to further injury. LP involves a process of electron transfer from membrane lipids to free radicals, resulting in cell damage, and most often affects the PUFAs. These compounds contain multiple double bonds and highly reactive hydrogens. The reaction is comprised of three major steps, namely initiation, propagation, and scission. The products of these reactions include lipid peroxyl radical, lipid hydroperoxide, and a peroxyl-fatty acid radical. The final step of the reaction, scission, leads to the formation of neurotoxic aldehydes such as 4-hydroxynonenal (4-HNE) and 2-propenal (acrolein). 4-HNE produces neurotoxicity via binding to basic amino acids, including lysine or histidine, as well as sulfhydryl-containing cysteine residues [8].

### 2.6. Glutathione Peroxidase

Glutathione peroxidase (GPx) and glutathione reductase (GR) are intracellular antioxidant enzymes. GPx-catalyzed oxidation of glutathione (GSH) produces its oxidized form (GSSG), while GR catalyzes the recycling of GSSG to GSH. Enzymes that remove superoxide and H_2_O_2_ protect the cell against oxidative stress. Catalase and GPx, in conjunction with SOD, are the major defense enzymes against superoxide radicals [50]. Due to oxidative stress, the function and transport of mitochondria to synaptic regions are impaired, leading to synaptic dysfunction and neurodegeneration [8,32,51].

### 2.7. Malondialdehyde

ROS-induced damage is defined by the onset of LP and is reflected by malondialdehyde (MDA) levels. MDA reacts with several functional groups on proteins, lipoproteins, and DNA. MDA levels have been measured to be elevated 1 min post-trauma, with a progressive increase to the maximum in the second hour, and persisting up to 48 h post-trauma [8].

### 2.8. Glutamate

TBI involves an excitotoxic physiological cascade, marked by excessive glutamate release. Glutamate plays a central role in various neurocognitive functions including cognition, memory, and learning [8,33]. Glutamate also plays a key role in brain development, neuronal signalling leading to cellular survival and differentiation, and the formation and elimination of synapses. TBI results in the excessive release of extracellular glutamate, in turn leading to the overstimulation of metabotropic and ionotropic receptors, which can ultimately lead to neuronal cell death. This process leads to the release of calcium (Ca^2+^) from internal stores, and in turn, the activation of *N*-methyl-D-aspartate (NMDA) receptor, 2-amino-3-(3-hydroxy-5-methylisoxazaol-4-yl)proprionic acid receptor (AMPAR), and kainite receptor. AMPAR receptor activation leads to the entry of Na^+^ into the cell, depolarization, and subsequent cellular edema. NMDA receptor activation also leads to the entry of Na^+^ and Ca^2+^ into the cell and subsequent membrane depolarization, and the induction of Ca^2+^-dependent enzymes, including calpains, protein kinases, endonucleases, and NOS [8,52].

### 2.9. N-acetylaspartate

*N*-acetylaspartate (NAA) is the acetylated form of aspartate and is ubiquitous in the central nervous system (CNS). NAA is a robust imaging biomarker in brain H-MRS studies. Using whole-brain NAA measurements by High-Performance Liquid Chromatography (HPLC), the levels of NAA have been correlated with the degree of injury [53]. Utilizing MR spectroscopy, NAA can be used to quantify neuronal damage and has been correlated with post-TBI neurocognitive sequelae [8,35,36].

### 2.10. Nitric Oxide Synthase

Nitric oxide synthases are a group of enzymes that facilitate the production of NO from *L*-arginine. TBI leads to the activation of NOS and, in turn, the formation of NO in the brain [54]. NO has been measured in the cerebrospinal fluid (CSF) of TBI patients [55], and the final products of NO and nitrate/nitrite levels have been correlated with TBI severity [56]. NO can be produced by three NOS isoforms: NOS1 (neuronal) and NOS3 (endothelial), which are Ca^2+^ dependent enzymes, and NOS2 (inducible), which leads to high levels of NO production in inflammatory situations. The role of NOS in TBI has been previously reported [45,57]. Non-selective NOS inhibitors given post-TBI, such as *L*-NAME and 7-nitroindazole, have been shown to reduce neurological deficits [37,38].

## 3. Oxidative Stress and Synapse

Oxidative stress has been reported in both clinical and preclinical TBI studies, and has been found to persist up to six months post-injury [4,58,59,60,61]. Synaptic alterations play a key role in the convergence of glutamate toxicity and oxidative stress. NMDA receptor activation and the influx of Ca^2+^ lead to ROS and RNS generation [62]. Postsynaptic density protein 95 (PSD95) plays a fundamental role in coupling PKCα to NMDA receptors. It also interacts with nitric oxide synthase (nNOS), leading to nitric oxide generation [61,63], which contributes to both inflammation and oxidative stress [8]. The physiological coupling of nNOS and PSD95 is foundational in oxidative stress. Disrupting PSD95 has been shown to be neuroprotective in a preclinical model of TBI [64]. Multiple prior preclinical TBI studies using a systems biology approach have suggested the potential role of nNOS and PSD95 as TBI biomarkers [65]. Prior preclinical TBI studies by Ansari et al. have examined the relationship between oxidative stress and synaptic density alterations in hippocampal and cortical areas [19,66].

A longitudinal analysis of post-TBI tissue demonstrated evidence of increased carbonylation, 3-nitrotyrosine, and acrolein, markers of oxidative stress. The increase in oxidative stress markers proceeded to a decrease in PSD95 level, at 24 h and 48 h time points. The temporal association suggests the potential role of oxidative stress in synaptopathy. There is a pronounced release of glutamate post-TBI, as reported in both clinical and preclinical studies [34,67,68,69]. The rise of glutamate is thought to be due to multiple potential etiologies, which include the extravasation of blood into the parenchymal space [70], the release of structural amino acids via newly developed micropores post-injury [67], dysregulation of astrocytic glutamate transporters GLT-1 and GLAS leading to glutamate uptake dysfunction, and increased synaptic release and resultant overflow from the synaptic terminal [71]. The increased glutamate post0injury has a profound effect on excitotoxicity and resultant synaptic alterations.

Complexins are 15-16 kDa cytosolic proteins that compete with the a-SNAP protein for binding with SNAP receptors (SNARE) in a rapid anti-parallel fashion in the presynaptic terminal [72]. The localization of complexin I in the inhibitory neuron and complexin II in the excitatory neurons in the cerebellum and hippocampus has provided a means of assessing excitatory and inhibitory pathways [73]. Yi et al. were able to demonstrate, using a lateral fluid percussion (FP) injury model in mice, that TBI leads to dynamic changes in complexin I and complexin II levels, and the aforementioned changes are associated with oxidative stress [71].

In another preclinical study by Ansari et al. using a unilateral moderated controlled contusion model of TBI in rats, it was shown that PSD-95, a post-synaptic scaffolding protein, followed a time-dependent decrease, observed at 24 h and perpetuating to 96 h post-injury. Similarly, synapsin-I, a major presynaptic protein, also followed a time-dependent change, with a significant change at 24 h and a continued decline after 96 h post-injury. Synapse-associated protein 97 (SAP-97), a membrane-associated phosphor-protein that interacts with a variety of different AMPA-type glutamate receptors, was shown to decline post-injury, with significant changes at 96 h post-injury [66].

The hippocampus, in charge of episodic memory, is another vulnerable locus of TBI injury due to its intrinsic connectivity and prevalence of high density of excitatory amino acid (glutamate) receptors [69,74]. In an accompanying study by Ansari et al., it was shown that the levels of presynaptic marker protein, synapsin I, SAP-97, and PSD-95 are significantly diminished post-TBI and remain decreased 96 h post-TBI [19].

## 4. Oxidative Stress and Inflammation

There are a growing number of studies highlighting the roles of cytokine, chemokines, and growth factors in the pathophysiology of TBI. In the acute post-injury period, there is a large production of cytokines, including IL-1β and tumor necrosis factor-α (TNF-α), as well as transforming growth factor-beta (TGF-β), which in turn enhances the brain trauma condition with increased oxidative stress and matrix metalloproteineases (MMPs) [12,75,76,77]. The posttraumatic inflammatory cascade leads to BBB dysfunction and an influx of inflammatory cells from the blood to the brain [78].

IL-1β triggers inflammatory reactions, leading to BBB disruption, edema, and neuronal loss [79,80,81,82]. Heightened IL-1β levels have been reported within the CSF and brain tissue in both humans and experimental animals [12,83,84]. Intracerebroventricular infusion of IL-1β-neutralizing antibody has been shown to reduce cerebral edema, neuronal loss, and lead to cognitive improvement following TBI in mice [85]. TNF-α plays key roles in neuronal development, cell survival, synaptic plasticity, and CNS ionic homeostasis. TNF-α has been measured in the CSF and serum of TBI patients. The TNF-α increase has also been associated with the loss of sensorimotor and cognitive functions in TBI models [86,87,88,89]. Multiple in vivo studies have reported an increased level of TGF-β within the CSF in multiple CNS disorders including TBI, which can be detected within days post-injury [90,91]. High levels of TGF-β within CSF could be due to damaged BBB, and its inhibition has been shown to prevent epilepsy post-injury [92].

Nuclear factor-erythroid-2-related factor (Nrf)-2 is a pleiotropic transcription factor and a master regulator of anti-oxidant defense response. It plays a major part in protecting cells from cytotoxic/oxidative damage [93,94]. The role of Nrf-2 as a protective agent in TBI-induced oxidation and the neuroinflammation cascade is increasingly being investigated [95]. Under basal conditions, Kelch-like ECH-associated protein 1 (KEAP1) binds Nrf2 and sequesters it in the cytoplasm. Post-injury, the increased ROS modifies cysteine residues in KEAP1 and decreases its affinity for Nrf2, and Nrf2 subsequently translocates into the nucleus. It subsequently combines with antioxidant response element (ARE) sequences, leading to activation of the Nrf2/ARE pathway and the gene expression of phase II detoxifying enzymes and antioxidases, such as heme oxygenase 1 (HO-1), SOD-1, and glutathione peroxidase 1 (GPx1). These enzymes protect cells from oxidative stress and a broad range of other toxins [14,96]. Using a preclinical TBI model, Cordaro et al. examined a new compound, Hidrox^®^, and demonstrated the modulation of oxidative stress and neuroinflammation, partly via reducing the activation of the nuclear factor kappa-light-chain-enhancer of activated B cells (NF-kB) pathway and associated cytokine activation [14]. TBI leads to a cascade of inflammatory processes, involving NF- kB, which contributes to further neuronal damage and delays in functional recovery [94]. TBI animals treated with Hidrox^®^ had significantly increased IkB-α levels in the cytosol and reduced NFkB expression in the nucleus [14].

## 5. Treatment Options of Oxidative Stress

There are a variety of preclinical studies focused on the utility of antioxidants in TBI. However, there are multiple challenges that make the clinical interpretation of such studies exceedingly more challenging, and these include the large variability in preclinical TBI models, the antioxidant tested in the study, the time interval of the antioxidant utilized, dosage, and the route of drug administration, as well as the range of neurocognitive parameters used [97].

Antioxidants function to robustly reduce the power of various oxidants, including ROS and RNS [97]. They can be generally divided into hydrophilic (water-soluble) and hydrophobic (fat-soluble) antioxidants. With the exception of reduced glutathione (GSH), uric acid, and coenzyme Q_10_, the primary sources of antioxidants are exogenous dietary sources [98]. Therefore, the supplementation of nutraceuticals is of key importance to provide an adequate supply of antioxidant protection [99].

Although the importance of the antioxidant regimen in TBI treatment is increasingly being recognized, there is no robust evidence for its efficacy in a clinical setting [97,100]. The reader is referred to an excellent review article written by Di Pietro et al. on this topic [97]. In Table 2, we summarize selected low-molecular-weight antioxidants that have been published in both preclinical and clinical TBI studies.

Oxidative stress is a result of disequilibrium between free radical formation and the antioxidant system. CSF MDA levels can rise within a 2–3 h window post-TBI and remain elevated 7 days post-injury. Moreover, SOD activity has been shown to diminish 24 h post-TBI, and remain so 7 days post severe TBI [8,101]. Antioxidant treatment focused on diminishing oxidative stress levels has been shown to reduce post-TBI neurological deficits, cerebral edema, and brain lesion volume [8]. Various antioxidant regimens have been studied, including polyethylene glycol (PEG)-conjugated SOD (PEG-SOD) [102], melatonin mimicking GPx [103], OPC-14117 (7-hydroxy-1-[4-(3-methoxyphenyl)-1-piperazinyl] [104], acetylamino-2, 2,4, 6-tetramethylindan (a superoxide anion scavenger), and phenyl-tert-butylnitrone (a free radical scavenger) [105], as well as mesylate tirilazad (an LP inhibitor) [28]. L-NAME treatment has been shown to decrease nitrotyrosine levels post-TBI, supporting the role of NO in PN formation. Furthermore, PN scavengers such as penicillamine, penicillamine methylester, and tempol have shown beneficial treatment effects [28].

### 5.1. Ascorbic Acid (Vitamin C)

Ascorbic Acid (AA) is a highly abundant water-soluble antioxidant and a key cofactor in multiple important enzymatic reactions [106]. As a robust reducing agent, it reacts with a range of ROS and RNS. AA deficiency is linked to debilitating connective tissue disease [107]. AA is not endogenously produced in the body, and exogenous dietary sources are required for the supply needed [108]. AA absorption occurs via SVCT1 Na^+^-AA transporter activity [109]. It has been reported that the levels of AA decrease dramatically after experimental TBI [110,111,112], and the changes in AA levels are correlated with the TBI severity [113]. In a preclinical study, using closed-head TBI, different doses of AA administered for 2 weeks, either alone or in addition to tocopherol, were shown to reduce the mortality rate, decrease melanoaldehyde (MDA) levels, and increase tissue superoxide dismutase levels [114]. A double-blind controlled clinical trial study by Razmkon et al. reporting on 100 TBI patients showed that a high dose of AA treatment led to the diminished progression of cortical edema [114].

### 5.2. N-Acetyl-Cysteine (NAC)

Reduced glutathione (GSH) has a fundamental role in various detoxifying reactions, which also includes scavenging ROS and RND, as well as the elimination of xenobiotics [115]. GSH is a low-molecular-weight antioxidant that mammals can endogenously synthesize. GSH is one of the main reducing agents for protein thiol groups and can also modulate inflammatory responses [116]. GSH is the cofactor for the enzymes GSH peroxidase (GPx) and glutathione-S-transferases. The reduction of oxidized GSH (GSSG) is performed by the NADPH-dependent enzyme GSH reductase (GR) [97]. GSH also binds to glutamate NMDA and AMPA receptors and can function as a neuromodulator [117]. A portion of the total cellular GSH content is in the form of *S*-nitroglutathione (GSNO), produced by a reaction between NO and GSH. GSNO is also a source of NO, another key antioxidant in lipid peroxidation reactions [19]. Prior preclinical studies have reported a decrease in GSH levels following TBI. There is a significant decrease in brain GSH levels following TBI, and more specifically, a decrease in GSH/GSSG levels has been reported in the subacute period post-TBI [20]. There is also a reported negative correlation between the degree of injury and rat brain GSH levels [110].

It has been shown in a preclinical study that i.p. injection of γ-glutamylcysteine ethyl ester (GCEE) in the post-injury period leads to a significant reduction of oxidative/nitrosative damage, highlighting the potential protective role of increasing GSH levels post-injury [118]. The therapeutic potential of *N*-acetylcysteine (NAC) is increasingly being recognized in both preclinical and clinical studies because of its direct role as a ROS and RNS scavenging agent or cysteine availability and GSH synthesis [119]. In a preclinical model of controlled cortical impact, NAC administration 15 min post-injury led to diminished MDA formation, increased SOD and GPx, and was found to be neuroprotective. Furthermore, NAC in combination with minocycline (MINO), given 12 h post-injury, prevented the TBI-induced reduction in the expression of oligodendrocyte markers CC1, 2′,3′-cyclic-nucleotide 3′-phosphodiesterase, and phospholipid protein in the subacute period post close head injury (CHI) [120]. An amidated form of NAC, namely *N*-acetylcysteine amide (NACA), can potentially provide more blood–brain barrier permeability, and potentially more robust therapeutic efficacy [121]. Complexin I and complexin II proteins, a family of cytosolic proteins and measures of inhibitory and excitatory synaptic activities, respectively, were shown to be completely blocked by *N*-acetylcysteine (NAC) administration 5 min post-injury in a preclinical study by Yi et al. [71]. Neuronal loss was also shown to be reduced in the injured hemisphere, post-TBI NAC treatment.

### 5.3. Flavonoids

Flavonoids are structurally identified by one phenyl ring fused with an oxygen-containing heterocyclic ring and a second phenyl ring. These compounds are ubiquitous in plants and fungi [97,122]. There are multiple subtypes of flavonoids, which include anthocyanidins, anthoxanthins, flavanones, flavanonols, and flavans. Due to their negative oxido-reductive potential, flavonoids can function as robust antioxidants towards various ROS and RNS species [123]. Due to the vast number of potential flavonoid sources, systematic clinical studies have been more difficult to conduct. A small pilot study, comprised of 60 mild TBI patients, using oral enzogenol (1 g daily) for 6 weeks, resulted in significantly fewer self-reported cognitive deficits [123].

### 5.4. Resveratrol

Resveratrol, a mostly hydrophobic compound, is abundantly existent in the skin of red grapes. Red wines contain an ample amount of resveratrol and have been thought to be the protective agent in the so-called “French paradox”, pertaining to low cardiovascular diseases in the French population, in spite of their high fat consumption [124]. It is proposed to have anti-inflammatory benefits, as well as modulate mitochondrial functions [125]. In a preclinical study, involving a weight drop model of TBI, rats treated with resveratrol (100 mg/kg) given via i.p. injection were found to have lower levels of MDA, xanthine oxidase (XO), and NO and increased levels of GSH as well as diminished parenchymal damage [126]. In another pre-clinical study involving a CCI-TBI preclinical model, resveratrol was shown to increase cell survival by inhibiting GSK-3β-mediated autophagy and apoptosis [127]. As per another preclinical study, resveratrol was found to have anti-inflammatory properties, noted by decreased microglial activity in various brain regions as diminished cytokine levels such as IL-6 and IL-12 levels [128].

### 5.5. Alpha-Tocopherol (Vitamin E)

Tocopherols are fat-soluble compounds, ubiquitous in vegetable oils, with potent antioxidant properties [97]. They are comprised of four tocopherols and four tocotrienols [129]. They contain a central chromane ring, which has oxidoreductive capacity. The brain, although rich in easily oxidizable fat-soluble compounds, has the lowest content of tocopherols [97]. In a preclinical study, using a mild or severe model of TBI, i.p. administration of alpha-tocopherol (100 mg/kg) was shown to diminish brain levels of MDA produced by oxidative/nitrosative stress-induced lipid peroxidation [129]. In another study using focal moderate TBI, Sprague–Dawley rats received i.p. injection of alpha -tocopherols (600 mg/kg) prior to TBI induction. These rats performed better on neurocognitive tests and showed less evidence of edema, inflammation, and a higher degree of cerebral tissue regeneration based on Nogo-A and NgR levels [130]. In another preclinical study, rats either received a regular diet with 500 IU/kg of -tocopherols for 4 weeks, or without it, prior to receiving focal mild TBI, using a fluid percussion-based injury. The group fed the tocopherol-rich diet had enhanced levels of SOD, Sir2, and BDNF caused by TBI, as well as diminished motor function impairments [131]. There is a relative paucity of clinical studies using tocopherols. According to Razmkon et al. using four groups of TBI patients, IM administration of tocopherol (400 IU for 7 days) was associated with diminished mortality rates and enhanced clinical outcomes [114].

### 5.6. Coenzyme Q10

Coenzyme Q is a component of the mitochondrial chain of aerobic organisms. It contains a quinone ring, which in turn allows it to exist in fully oxidized, fully reduced, or semi-reduced forms [97]. Coenzyme Q10 (CoQ10) is found in the mitochondrial inner membrane, and actively participates in electron transfer [132]. There are a number of preclinical studies that have shown the efficacy of CoQ10 in the treatment of head trauma. In a preclinical Marmarou TBI model, CoQ10 (10 mg/Kg) was administered immediately after trauma, and after 24 h via gavage. The animals treated with CoQ10 had diminished MDA levels, vascular congestion, neuronal loss, nuclear pyknosis, nuclear hyperchromasia, cytoplasmic eosinophilia, and axonal edema, and increased cerebral SOD [133]. CoQ10 was also found to modify the cerebral expression of genes involved in bioenergetics and oxidative/nitrosative stress [134]. A similar preclinical study involving the administration of CoQ10 infusion before or 30 min after TBI led to diminished brain mitochondrial damage and apoptosis, as well as lower serum levels of serum glial fibrillary acidic protein (GFAP) and ubiquitin C-terminal hydrolase-L1 (UCH-L1) [135].

### 5.7. Carotenoids

Carotenoids are a group of pigments synthesized by plants, algae, and species of bacteria [97,136]. They are further subdivided into oxygen-containing carotenoids (xantophylls) and a class of oxygen-free carotenoids (carotenes). The use of different carotenoids as a potential therapeutic modality has been studied in depth in various neurodegenerative diseases [136]. In a preclinical model of TBI using astrocytes, astaxanthin diminished apoptosis via the inhibition of NKCC1 expression and reduced the expression of NF-κB-mediated pro-inflammatory factors [137]. In another study using the CCI preclinical model of TBI, astaxanthin of various doses (10, 25, 50, or 100 mg/kg body weight) administered 30 min post-injury decreased TBI-related brain tissue injury by inhibiting AQP4/NKCC1-mediated cerebral edema in a dose-dependent manner and was found to improve neurologic deficits and the BBB permeability [138]. In a separate study, involving a weight drop model of TBI, astaxanthin was found to reduce lesion size in the cortex and increase brain-derived neurotrophic factor (BDNF), growth-associated protein-43 (GAP-43), synapsin, and synaptophysin (SYP), supporting an induction of neuronal survival and plasticity, also evident by enhanced cognitive recovery [139]. Other carotenoids, including fucoxanthin, bexarotene, crocin, and lutein, have shown similar beneficial results in prior preclinical studies [137,140,141,142,143].

### 5.8. Omega-3 Fatty Acids

Omega-3 fatty acids belong to a group of polyunsaturated fatty acids (PUFA) that are subdivided into essential (linoleic acid) or semi-essential (eicosapentaenoic acid, docosahexaenoic acid) fatty acids for humans. Omega 3 fatty acids are not only foundational components of membrane phospholipids, but also precursors of key biologically active molecules (prostaglandins) [97]. Both docosahexaenoic acid (DHA) and eicosapentaenoic acid (EPA) are abundant in brain phospholipids, with DHA comprising 50% of the weight of neural tissue within the brain [144]. A number of preclinical studies have reported the benefits of omega-3-fatty acids and DHA in preclinical TBI models [97]. However, clinical studies, including a study performed by Matsouoka et al. using 53 TBI patients, with 52% of the participants with mild TBI, and involving the administration of 12 weeks of 1470 mg DHA and 147 mg EPA, did not confer any neurological (including measures of BDNF), cognitive, or affective benefits based on a multitude of neuropsychological batteries [145].

There is a synergistic reaction between dietary DHA and exercise. In a preclinical model using the mild fluid percussion injury (mFPI) model, it was shown that rats that underwent mFPI and were on a diet high in DHA (1.2% DHA), combined with 12 days of exercise, had elevated BDNF and p-TrkB levels, which are involved in synaptic plasticity and cognitive function. These levels were significantly different from the exercise-only group and regular diet-only group, respectively [146].

### 5.9. Pycnogenol

The neuroprotective effects of pycnogenol (PYC) have been explored in rodent models of TBI [147]. PYC is a combination of bioflavonoids, which are extracted from the bark of the French maritime pine tree (*Pinus maritima*) and have a robust capacity to scavenge free radicals. Ansari et al., using a well-established preclinical model of TBI, demonstrated the neuroprotective effects of PYC in TBI. PYC treatment was shown to reduce oxidative stress, and rescue both pre- and post-synaptic proteins in the cortex and the hippocampus. The level of neuroprotection was shown to be dose-dependent, with a therapeutic window extending up to 4 h post-trauma. However, there was a therapeutic time effect, with animals treated with PYC at 15 min demonstrating significantly lower levels of oxidative stress compared to a 2 or 4 h delay following injury within the cortex. However, the hippocampus did not show a significant time effect [147].

### 5.10. Phenelzine

Phenelzine (PZ) is an FDA-approved drug that functions as an MAO inhibitor. PZ has been shown to have aldehyde-scavenging properties and provide partial protection from LP-mediated oxidative injury [148,149,150]. In a preclinical study using a controlled cortical impact injury (CCI) model, PZ was administered subcutaneously (3–30 mg/kg) at different times post-injury. It was shown that PZ treatment preserved both synaptic and non-synaptic mitochondrial bioenergetics at 24 h, and this protection was partially extended to 72 h post-injury. PZ administration was also shown to significantly improve mitochondrial respiration, with a more robust response of synaptic mitochondria compared to the non-synaptic mitochondria [148].

**Table 2 ijms-23-13000-t002:** Outlines a number of reported therapeutic agents used in treatment of TBI-induced oxidative stress.

Therapeutic Agent	Mechanism	References
Ascorbic Acid (Vitamin C)	As a robust reducing agent, it reacts with a range of ROS and RNS.	Castiglione et al. 2018 [108] Parker et al. 2015 [112]
N-acetylcysteine (NAC)	direct role as ROS and RNS scavenging agent, or increasing cysteine availability and GSH synthesis, extensively studied in preclinical and clinical settings	Limongi et al. 2019 [115] Koza et al. 2019 [119]
Flavonoids	Given their negative oxido-reductive potential, flavonoids can function as robust anti-oxidants towards various ROS and RNS species	Izzi et al. 2012 [122] Theadom et al. 2013 [123]
Resveratrol	Has both anti-inflammatory benefits, as well as modulation of mitochondrial functions	Irias-Mata et al. 2017 [128] Gatson et al. 2013 [127]
a-Tocopherol (Vitamin E)	Fat soluble compounds, ubiquitous in vegetable oils, with potent anti-oxidant properties	Di Pietro et al. 2020 [97] Razmkon et al. 2011 [114]
Coenzyme Q10	component of the mitochondrial chain aerobic organisms with efficacy in TBI treatment in a few preclinical studies	Pierce et al. 2017 [134] Pierce et al. 2018 [135]
Carotenoids	group of pigments synthesized by plants, algae, and species of bacteria, and in preclinical studies have bene shown to diminish apoptosis, and reduced expression of NF-κB-based inflammation	Zhang et al. 2017 [137] Ji et al. 2013 [139]
Omega-3 Fatty Acids	group of polyunsaturated fatty acids (PUFA), that are subdivided into essential or semi-essential for humans, and benefits of omega-3-fatty acids and DHA in preclinical TBI models	Matsuoka et al. 2015 [145] Wu et al. 2013 [146]
Pycnogenol (PYC)	combination of bioflavonoids with robust capacity to scavenge free radicals	Ansari et al. 2013 [147]
Phenelzine	FDA-approved drug, which functions as an MAO inhibitor and has been shown to have aldehyde scavenging properties, and provide partial protection from LP-mediated oxidative injury	Hill et al. 2020 [148] Cebak et al. 2017 [149]

## 6. Conclusions

TBI remains a leading cause of morbidity and disability globally as our collective understanding of the disorder continues to evolve. Oxidative stress is a key component of the secondary injury cascade that can often be perpetuating and chronic. As there are currently no FDA-approved regimens for TBI treatment, alternative therapies such as the use of antioxidants described here become a viable treatment option for consideration in the clinical setting. TBI is inherently a heterogeneous disease due to variance in both the mechanism and severity of injuries. Hence, a range of treatment options will be needed to meet the challenge. The preclinical TBI models, although salient from a mechanistic point of view, can be challenging to translate into a real clinical setting. In addition, there is a great deal of temporal variation between the time of injury and clinical assessment. The use of antioxidants may provide a relatively safe option to offer in an acute treatment setting, which in turn would alter the trajectory of the disease process. Future studies, both on preclinical and clinical levels, can be focused on TBI progression in terms of oxidative stress and the potential efficacy of antioxidant treatment on multiple different time scales.

## Figures and Tables

**Figure 1 ijms-23-13000-f001:**
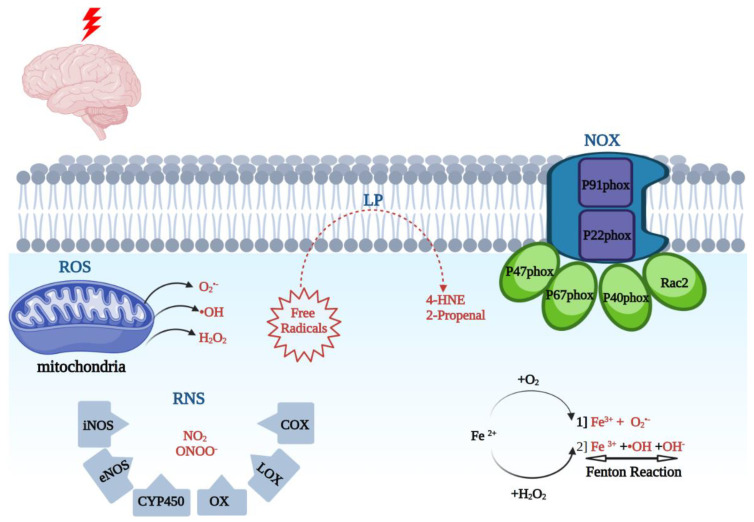
Schematic illustration of the main cellular components involved in the post-TBI oxidative stress process. The NAPDH oxidase (NOX) with its two membrane subunits, P91phox and P22phox, and cytoplasmic components, P47phox, P67phox, P40phox, and Rac2, is a key enzymatic unit involved in this process. The reactive oxygen species (ROS), comprised of superoxide (O2●−), hydrogen peroxide (H_2_O_2_), and the hydroxyl radical (●OH), are products of 1-electron reduction of oxygen. The reactive nitrogen species (RNS), including various nitric oxide compounds, such as peroxynitrite (ONOO^-^) and nitrogen dioxide (NO_2_), are the products of multiple enzymes including nitric oxide synthase (iNOS), endothelial nitric oxide synthase (eNOS), cytochrome P450 (CYP450), cyclooxygenase (COX), lipoxygenase (LOX), and xanthine oxidase (XO). Free iron (Fe^2+^), an abundant ion within the CNS, participates in two main reactions, including the Fenton reaction, leading to the production of Fe^3+^, O2●−, ●OH +, and OH^−^. The phenomena of lipid peroxidation (LP) involves electron transfer from membrane lipids to free radicals, involving a three-step process, and ultimate production of 4-HNE and 2-propenal (acrolein). The free radicals are depicted as red vs. the enzymes and proteins are in black. Images created with BioRender.com.

**Table 1 ijms-23-13000-t001:** Outlines the main pathological agents involved in TBI oxidative stress, and their corresponding proposed mechanisms.

Pathological Agent	Mechanism	References
Reactive Oxygen Species (ROS)	ROS includes superoxide (O2●−), hydrogen peroxide (H_2_O_2_), and the hydroxyl radical (●OH), produced by mitochondria and product of cellular oxidative metabolism	Eastman et al. 2020 [1] Pisochi et al. 2015 [24]
Reactive Nitrogen Species (RNS)	RNS Includes nitric oxide compounds, such as peroxynitrie (ONOO^−^), and nitrogen dioxide (NO_2_), and have longer half-lives than ROS	Balazy et al. 2003 [25] Brandes et al. 2010 [26] Uttara et al. 2009 [27]
Peroxynitrite (PN)	PN can also react with carbon dioxide (CO2), and form nitoperoxocarbonate (ONOOCO2) & subsequent cellular damage via transfer of electron from hydrogen atom bound to allylic carbon in polyunsaturated fatty acids (PUFAs)	Hall et al. 2012 [28] Hall et al. 2004 [21]
Iron and Hydroxyl Radicals	Post TBI, the PH within injured areas are lowered, in turn leading to release of iron from its storage sites with an additional source of catalytically active iron being hemoglobin, in turn responsible for hemoglobin mediated oxidative damage	Daglas et al. 2018 [29] Hatakeyama et al. 2013 [30] Zeng et al. 2018 [31]
Lipid Peroxidation (LP)	LP has been associated with loss of cell membrane integrity, increased membrane permeability, diminished activity in membrane ATPase, in turn leading to further injury	Cornelius et al. 2013 [8]
Glutathione Peroxidase (GPx)	GPx reactions lead to oxidation of glutathione (GSH) to oxidized form (GSSG)	Kushner et al. 1998 [32]
Malondialdehyde (MDA)	MDA reacts with several functional groups on proteins, lipoproteins and DNA, leading to further oxidative damage	Cornelius et al. 2013 [8]
Glutamate	TBI involves excessive release of extracellular glutamate, overstimulation of metabotropic and ionotropic receptors, and neuronal cell death	Riedel at al. 2003 [33] Yi et al. 2006 [34]
N-acetylaspartate (NAA)	N-acetylaspartate (NAA) is the acetylated form of aspartate, ubiquitous in CNS and a robust injury marker	Signoretti et al. 2001 [35] Tavazzi et al. 2000 [36]
Nitric oxide synthase (NOS)	TBI leads to activation of NOS, in turn the formation of NO in the brain	Besson et al. 2009 [37] Ishida et al. 2001 [38]

## Data Availability

Not applicable.

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
