# Peer review of "Oxidative Stress in Traumatic Brain Injury"

_ijms, 2022, doi:10.3390/ijms232113000_

Round 1
Reviewer 1 Report
This review summarized the role of oxidative stress in TBI and possible therapeutic approach
The topic is of interest; but minor points need to be addressed by authors
1) There are many grammatical and spelling errors throughout the manuscript. Please, have the paper carefully edited and corrected.
2) The authors should describe the different experimental models of TBI doi: 10.3390/antiox9040297. doi: 10.1139/bcb-2016-0160. doi: 10.1080/00207454.2017.1380008.
3) The authors should also more report the role of nrf-2 and NF-kB transcription factors on TBI and the effects of new compounds with antioxidant effect obtained by the modulation of nrf2 or nf-kB pathways in animal models of TBI such as doi: 10.3390/antiox10050818. doi: 10.3390/antiox10060898, https://doi.org/10.1186/s12974-020-01863-, doi. https://doi.org/10.1177/0271678X17738701
Author Response
This review summarized the role of oxidative stress in TBI and possible therapeutic approach
The topic is of interest; but minor points need to be addressed by authors
I am grateful for the thoughtful comments and recommendations of the reviewers. I have made a concerted effort to address the raised issues, and I have made corresponding major changes to the original manuscript with the aim of improving the quality and clarity of the document.
- There are many grammatical and spelling errors throughout the manuscript. Please, have the paper carefully edited and corrected.
>> I have made significant revisions including correction of the typos to the text.
2) The authors should describe the different experimental models of TBI doi: 10.3390/antiox9040297. doi: 10.1139/bcb-2016-0160. doi: 10.1080/00207454.2017.1380008.
>> This reviewer’s comment is definitely appreciated, and I have made additions to the text.
- The authors should also more report the role of nrf-2 and NF-kB transcription factors on TBI and the effects of new compounds with antioxidant effect obtained by the modulation of nrf2 or nf-kB pathways in animal models of TBI such as doi: 10.3390/antiox10050818. doi: 10.3390/antiox10060898,https://doi.org/10.1186/s12974-020-01863-, doi. https://doi.org/10.1177/0271678X17738701
>> The corresponding material regarding nrf2 or nf-kB pathways has been added to the oxidative stress and inflammation subsection.
Reviewer 2 Report
Comments and suggestions
1. Abstract should be focused according to your title, please check and revise.
2. Make a graphical abstract for understanding the readers.
3. Introduction section needs to be rewritten.
4. Free Iron (Fe 2+ ), an abundant ion within the CNS, participates in two 61 main reactions leading, including the Fenton reaction, leading to production of Fe 3+ , O2●-, ●OH + 62 and OH- . not written correctly
5. Table 1 is not cited in the text, is it necessary red color the left column of the table?
6. Section 2 only describes oxidative stress and no connection with Traumatic Brain Injury
7. Lot of formatting and typos errors throughout the manuscript.
8. Most of the references used are very old.
I am worried there is a connection missing between the content and title. Even with major revisions, it's difficult to improve this manuscript.
However, all the best, and expect good revision.
Author Response
Reviewer #2
- Abstract should be focused according to your title, please check and revise.
>> I have made a number of edits to the abstract, with aim of further clarification and specificity.
- Make a graphical abstract for understanding the readers.
>> The graphical abstract has been added.
- Introduction section needs to be rewritten.
>> I have made major revisions to the introduction section.
- Free Iron (Fe 2+ ), an abundant ion within the CNS, participates in two main reactions leading, including the Fenton reaction, leading to production of Fe 3+ , O2-, ●OH + 62 and OH- . not written correctly
>> This was corrected and edited in the text.
- Table 1 is not cited in the text, is it necessary red color the left column of the table?
>> This has been cited in the “Section II: Pathophysiology” section of the manuscript.
- Section 2 only describes oxidative stress and no connection with Traumatic Brain Injury
>> This is further elaborated in section II.
- Lot of formatting and typos errors throughout the manuscript.
>> I appreciate this comment, and major revisions including typos have been made.
- Most of the references used are very old.
>> This is an extremely apt point, and going through the manuscript, I wholeheartedly agreed! I have made a significant revision of the bibliography accordingly.
I am worried there is a connection missing between the content and title. Even with major revisions, it's difficult to improve this manuscript.
However, all the best, and expect good revision.
Round 2
Reviewer 2 Report
Thanks for improving your manuscript. But still needs improvement for consideration in this journal.
Comments and suggestions
1. Introductions section needs more updates. Please write your aim of review at the end of the introduction.
2. Figure 1. Schematic illustration of the main cellular components involved in the post TBI oxidative stress process. The 77 NAPDH oxidase (NOX) with its two membrane subunits, P91phox and P22phox, and cytoplasmic components, P47phox, 78 P67phox, P40phox and Rac2 is a key enzymatic unit involved with this process. The reactive oxygen species (ROS), 79 comprised of superoxide (O2●-), hydrogen peroxide (H2O2) and the hydroxyl radical (●OH), are 80 products of 1-electron reduction of oxygen. The reactive nitrogen species (RNS), including vari- 81 ous nitric oxide compounds, such as peroxynitrie (ONOO- ), and nitrogen dioxide (NO2), are the 82 products of multiple enzymes including nitric oxide synthase (iNOS), endothelial nitric oxide 83 synthase (eNOS), cytochrome P450 (CYP450), cyclooxygenase (COX), lipoxygenase (LOX), and 84 xanthine oxidase (XO). Free Iron (Fe 2+ ), an abundant ion within the CNS, participates in two 85 main reactions leading, including the Fenton reaction, leading to production of Fe 3+ , O2●-, ●OH + 86 and OH- . . The phenomena of lipid peroxidation (LP), involves electron transfer from membrane lipids to free radicals, 87 involving a three-step process, and ultimate production of 4-HNE and 2-Propenal (acrolein). Images created with Bio- 88 Render.com. Please check equation has been written correctly. Also check font size.
3. Section 2.2, 2.3, 2.4, etc should be expanded with relevant information according your study.
4. Table 1& 2 make one more column for references.
5. References has not been formatted accordingly to journal guideline that should be checked carefully.
Author Response
Please refer to the attached file with individual responses to reviewer.